# Technical Evaluation of qPCR Multiplex Assays for the Detection of *Ixodes ricinus*-Borne Pathogens

**DOI:** 10.3390/microorganisms10112222

**Published:** 2022-11-10

**Authors:** Tal Azagi, B. J. A. Hoeve-Bakker, Mark Jonker, Jeroen H. Roelfsema, Hein Sprong, Karen Kerkhof

**Affiliations:** Centre for Infectious Disease Control, National Institute for Public Health and the Environment, P.O. Box 1, 3720 BA Bilthoven, The Netherlands

**Keywords:** *Anaplasma phagocytophilum*, *Candidatus Neoehrlichia mikurensis*, *Spiroplasma ixodetis*, *Babesia microti*, *Babesia divergens*, *Rickettsia helvetica*, *Rickettsia* stenos group, *Bartonella* spp., multiplex qPCR, Taqman

## Abstract

Background: The extent to which infections with *Ixodes ricinus*-borne pathogens (TBPs), other than *Borrelia burgdorferi* s. l. and tick-borne encephalitis virus (TBEV), cause disease in humans remains unclear. One of the reasons is that adequate diagnostic modalities are lacking in routine or research settings. Methods: We evaluated the analytical specificity, sensitivity and robustness of qPCR assays for the detection of *Anaplasma phagocytophilum*, *Neoehrlichia mikurensis*, *Spiroplasma ixodetis*, several *Babesia* species and Spotted Fever *Rickettsia* species as well as *Bartonella* species in human samples. Results: The qPCRs were found to perform well, given the difficulties of dealing with microorganisms for which confirmed patient materials are scarce or non-existent, a hurdle that was partially overcome by using synthetic controls. Spiking blood samples with the tested microorganisms showed that the detection of the TBPs was not inhibited by the presence of blood. The acceptable sensitivity when multiplexing the different pathogens, the good inter-assay variability and the absence of cross-reactivity make them potentially suitable as human diagnostics. Conclusions: The qPCRs evaluated in this study are technically suitable for the laboratory diagnostic assessment of clinical samples for infection with tick-borne pathogens. However, clinical validation and independent confirmation are still needed, pending the availability of sufficient human samples for testing in different laboratories.

## 1. Introduction

*Ixodes ricinus* is a hard-bodied tick species that transmits a plethora of human pathogens. Infections with the *Ixodes ricinus*-borne pathogens (TBPs) *Borrelia burgdorferi* sensu lato, tick-borne encephalitis virus (TBEV), *Anaplasma phagocytophilum* [1], *Babesia divergens* [2], *Babesia microtii* [3], *Babesia venatorum* [2], *Borrelia miyamotoi* [4], *Neoehrlichia mikurensis* [5], *Spiroplasma ixodetis* [6], *Rickettsia helvetica* [7] and spotted fever group *Rickettsia* stenos [8] have all been associated with human disease [6,9,10].

Especially, *Borrelia burgdorferi* s. l., which causes Lyme Borreliosis, is the most common tick-borne disease in Europe [11,12,13], with more than 200,000 cases per year in western Europe [11]. Infections with TBEV can cause a serious neurological disease. TBEV is the most frequent arboviral disease in Europe, with 2000–3500 human cases each year. Other tick-borne diseases (TBDs) are far less frequently reported in the general population [9,14,15]. Several studies have shown that there is substantial exposure of humans to these TBPs through tick bites, such as *B. miyamotoi*, *A. phagocytophilum*, *Babesia* spp., *B. microti*, *N. mikurensis*, *R. helvetica* and *S. ixodetis* [16,17,18]. For some of these TBPs, the pathogenicity remains questionable, and only few well described cases exist [9]. Over the last decades, only two autochthonous anaplasmosis cases, two *B. miyamotoi* cases, but no cases of neoehrlichiosis or human babesiosis have been reported in the Netherlands [19]. Lack of awareness, case definitions, laboratory diagnostics, as well as a non-characteristic clinical presentation are among the reasons why these other TBDs often go undiagnosed [12,20].

Adequately diagnosing cases caused by TBPs, except for *B. burgdorferi* s.l. and TBEV, is challenging, as detection requires the use of standardised diagnostic modalities in common practice. Currently, diagnostic assays for TBPs, such as *N. mikurensis* and *R. helvetica*, are only available in specialised laboratories or research settings [10].

In a clinical setting, real-time polymerase chain reaction (qPCR) is widely used as a time- and cost-effective specific method for TBP detection during the acute phase of infection [4,21]. As the infectious dose for TBPs may be low, a low detection limit is required to detect the pathogen, especially in the early stage of the disease [22]. A thorough knowledge of the assay performance characteristics of the chosen detection method is the only way to understand the resolution and appropriate use of any results obtained.

In this study, the diagnostic performance of three multiplex in-house qPCR assays for (i) the simultaneous detection of *A. phagocytophilum*, Ca. *N. mikurensis* and *Babesia* spp., (ii) *R. helvetica*, *B. microti* and *Spiroplasma ixodetis* and (iii) spotted fever group *Rickettsia* stenos and *Bartonella* spp. were evaluated. These evaluated assays have previously been used in singleplex in various studies in the Netherlands where the prevalence of TBPs in both questing and fed *I. ricinus* [17,19,23], animal hosts [17,23], human populations at risk as well as the general population [9,16,24] has been assessed. The aim of this study was to perform a technical validation of these qPCRs in order to estimate their potential in human diagnostics. The sensitivity, specificity and robustness of each multiplex in-house qPCR assay was evaluated.

## 2. Materials and Methods

### 2.1. Clinical Samples

Patient material was collected from the biorepository of the Centre for Infectious Disease Control (CIb) at the National Institute for Public Health and the Environment (RIVM) between 2015 and 2017. The material varied in source and quantity depending on the disease manifestation and clinical presentation. Skin biopsies (epithelium) were received from suspected *Rickettsia* patients and blood from patients with a suspicion of the other TBPs. Total genomic DNA was isolated with the High Pure Kit from Roche (Almere, The Netherlands) according to the manufacturer’s instructions and stored at −20 °C.

### 2.2. Plasmids

For the optimisation of the PCR conditions and the technical evaluation of the qPCRs, plasmids were constructed containing a target sequence that slightly differs from wildlife sequences. Therefore, a positive control was placed on the plasmid, which gave the target sequence a different size and sequence in relation to the wildlife sequence. The plasmids were obtained from GenScript Biotech Corp. (Piscataway, NJ, USA), and the pUC57 vector (2710 bp) was used for all microorganisms (Table 1).

The plasmids were dissolved according to the protocol of GenScript Biotech Corp. Each of the lyophilized 4 μg plasmids was dissolved in Tris-EDTA (TE) to obtain a 0.01 μg/μL (*Anaplasma*, *Neoehrlichia*, *Rickettsia* spp., *R. helvetica*) or a 0.02 μg/μL (*Babesia* spp. 18S, *S. ixodetis*, *B. microti*, *Bartonella* spp.) stock solution.

The conversion between plasmid copy number and mass was calculated using the formula: m = n × (1.096 × 10^−21^)g/bp, where n = DNA size in base pairs, and m = mass in μg [25].

### 2.3. Primer Pairs and TaqMan Probes Used for the LightCycler qPCR Assays

For three multiplex qPCR assays for the detection of the different TBPs, the primer pairs and probes are listed in Table 2. The primers and probes were obtained from Biolegio, Nijmegen, The Netherlands.

### 2.4. qPCR

All qPCR reactions were carried out on a Roche LightCycler^®^ 480 qPCR System (Roche Diagnostics Nederland B.V, Almere, The Netherlands) in a final reaction volume of 20 μL containing 10 μL of SensifastTM (Bioline, London, UK), 5 μL of sample, 10 pmoles of each of the primers and 2.5 pmoles of probe. The initial denaturation was performed at 95 °C for 3 min. For the amplification, 45 cycles at 95 °C for 5 s and 60 °C for 30 s were used.

To minimize contamination and false-positive reactions, DNA extraction, PCR mix preparation, sample addition, and qPCR analyses were handled in separate air-locked dedicated labs.

### 2.5. Technical Evaluation of the qPCR Multiplex Assays

#### 2.5.1. Sensitivity of the qPCRs

The dynamic range, the coefficient of determination and the limit of detection (LOD) were determined for the eight assays individually by plotting average cycle threshold (CT) values from three individual runs against the log10 of the 10-fold serial dilutions of plasmid DNA (copy number/µL). The LOD is defined as the lowest concentration of TBP target at which 95% of the samples are positive. The LOD was established using Finney’s Probit Analysis. First, a rough estimation of the LOD was made using tenfold serial dilutions of the target DNA. Then, serial dilutions were made in triplicate, expecting outcomes close to 100% positive PCR results at the highest concentration and close to 0% positive results at the lowest concentration. Between these concentrations, there were eight concentrations leading to diminished percentages of the positive results. The conversion between plasmid copy number and mass was calculated as described above.

The assays’ overall efficiency (E) was estimated using the slope of the standard curve: E = ((10^−1/slope^) − 1) × 100, in which E (%) indicates the fraction of target molecules copied in one PCR cycle.

The coefficient of determination (R2) is the square of the Pearson’s correlation coefficient (r) that refers to how well the CT values correlate with the dilution series. An R2 > 0.98 is acceptable and indicates the consistency of the serial dilutions and pipetting errors.

As a positive human control sample was not available for every TBP, this experiment was repeated using spiked blood to determine the matrix effect. All data were compared using paired *t*-tests (*p* > 0.05) and the correlation coefficient.

#### 2.5.2. Evaluation of Specificity

The specificity of the primers/probes for the three multiplex qPCR assays (Table 2) was confirmed in silico by using the Basic Local Alignment Search Tool (BLAST; https://blast.ncbi.nlm.nih.gov/Blast.cgi (accessed on 12 October 2022)). Additionally, possible cross-reactivity with multiple related (non-target) microorganisms that cause fever in Western Europe was ruled out. For this purpose, nucleic acid extracts from pure cultures of *Plasmodium falciparum*, *Parvovirus B19*, *Treponema pallidum*, *Coxiella burnetii*, *Escherichia coli*, *Staphylococcus aureus*, *Streptococcus pneumoniae* and *Streptococcus pyogenes* were tested in 1:10 and 1:100 dilutions in all three multiplex qPCR assays. Lastly, to evaluate the reactivity of the plasmids used in each multiplex assay, they were tested in the other multiplex assays using 10^−2^ and 10^−3^ dilutions.

#### 2.5.3. Reproducibility and Repeatability

The critical values of the three multiplex qPCRs were evaluated on four different LightCycler^®^ 480 machines. Therefore, at each LightCycler, a qPCR run was performed in triplicate using a ten-fold serial dilution of each of the positive control plasmids (Table 1). The reproducibility (inter-assay variability) was analysed based on the standard deviation (SD) and the Coefficient of Variation (% CV) of the mean CT values. A good inter-assay reproducibility of all individual and multiplex qPCR outcomes is seen when the % CV is less than 15% [29].

To determine the repeatability for *A. phagocytophilum*, five EDTA-anticoagulated blood samples were compared between two laboratories (RIVM (Bilthoven, The Netherlands) and the Military Hospital Queen Astrid (MHQA, Brussels, Belgium)). Two out of the five EDTA blood samples were blindly spiked with *A. phagocytophilum*. Since no sufficient reference human blood was available for the other targets, the repeatability could not be determined.

## 3. Results

### 3.1. qPCR Assay Performance (Sensitivity, Robustness and Precision)

The plasmids for each qPCR TBP target were used individually to determine the efficiency of the assay by amplifying ten-fold serial dilutions from 10^10^ copies/µL to 10 copies/µL in triplicate. The slopes of every target ranged from −3.0 to −3.4 (Table 3). Using the slope from the linear equation that was generated from the standard curve, the overall efficiency for MPX I was calculated at 100%, while for MPX II 111% and for MPX III, it was 96%. The efficiencies of MPX I and MPX III fell within the acceptable efficiency range (between 90 and 110%; [29]). On the other hand, MPX II exceeded that acceptable range by 1%, due to the individual estimated efficiency of 114% for *S. ixodetis* (Table 3).

The dynamic range, the R2 and the LOD were determined for the eight targets individually. A wide dynamic range (from CT 16.63 to CT 45.28) was seen for each of the targets while maintaining amplification linearity (Table 3). The R2 generated from the linear equation ranged between 0.946 (*R. helvetica*) and 0.999 (*A. phagocytophilum* and *Babesia* spp. 18S). R2 > 0.980 is acceptable for well-designed qPCR assays and indicates the consistency of serial dilutions [29]. The LOD for the eight individual assays were close to each other and varied between approximately 10 and 100 copies of plasmid DNA per reaction (Table 3).

This experiment was repeated identically on negative whole blood spiked with the target DNA of each TBP and compared to the dH_2_O-diluted plasmids. The correlation coefficient R2 of both experiments ranged from 0.960 to 0.999 (Table 4). No significant differences were observed for each of the TBP targets, suggesting that the use of whole blood did not affect the assay results.

### 3.2. Specificity of the qPCRs

The BLAST analysis revealed no homology with the reported sequences from species other than the targeted microorganisms, indicating that the targets were highly specific. Additionally, the clinical specificity of each assay was evaluated by screening samples containing the common pathogens *Plasmodium falciparum*, *Parvovirus B19*, *Treponema pallidum*, *Coxiella burnetii*, *Escherichia coli*, *Staphylococcus aureus*, *Streptococcus pneumoniae* and *Streptococcus pyogenes*. The clinical specificity of all qPCR assay was 100%, as no detectable signal was found and no amplification occurred for any of the tested non-target microorganisms samples.

### 3.3. Reproducibility and Repeatability of the qPCRs

The inter-assay %CV for the CT-values was determined for each of the eight individual targets and ranged from 1.48% to 11.52% (Table 5). The average %CV for the multiplex qPCRs was 2.24% for MPX I, 3.23% for MPX II and 6.55% for MPX III.

For the *A. phagocytophilum* target, the observed repeatability between the two European laboratories was 100%. The MPX I qPCR assay results at RIVM showed a similar outcome compared to those of MHQA, Belgium. The CT values of EDTA 1 were 33.91 and 37.19, respectively, and the CT values of EDTA 4 were 20.89 and 22.26, respectively. No DNA was detected in the remaining EDTA samples 2, 3 and 5 at both laboratories.

## 4. Discussion

The continued geographic expansion of ticks harbouring human pathogens and the identification of emerging TBPs in humans, such as *N. mikurensis*, *B. miyamotoi*, *R. helvetica* and *S. ixodetis*, warrant a reconsideration of the “norms” for the diagnosis of TBDs [16,17,18]. The extent to which this growing array of emerging TBPs causes disease in humans remains unclear. One of the reasons is that adequate diagnostic modalities are lacking in routine or research settings, which is further complicated by the variable performance of the available diagnostic assays for the diagnosis of tick-borne infections [12,20]. In the case of emerging TBPs, the microbial load in the blood is often low, which makes the development of diagnostics more difficult. The use of qPCR as a diagnostic tool has become very popular, as it is a fast and highly sensitive and specific method. Additionally, the risk of carry-over contamination is reduced compared to conventional diagnostic methods [30]. Therefore, the aim of this study was to transform the qPCR assays currently employed in our laboratories for the detection of TBPs in ticks and non-human hosts into three multiplex qPCR assays and to assess their applicability as human diagnostics. This is, to our knowledge, the first multiplex qPCR in which the not regularly occurring pathogens *S. ixodetis* is also included.

This endeavour was performed within the limits of dealing with microorganisms for which confirmed patient materials are scarce or non-existent [10,19], a hurdle that was partially overcome by using synthetic controls for the technical evaluation of the assays. Using the multiplex qPCRs, as little as 10 to 79 copies per PCR reaction could be detected for each of the targeted TBPs, which is a 10 to 10,000 times higher sensitivity compared to that of the multiplex PCR for the detection of nine TBPs evaluated in a similar manner by Buchan et al. [31]. Likewise, the current study demonstrated that the detection of TBPs was not inhibited when DNA was isolated from human blood and that the qPCR results had a good inter-assay variability, suggesting this assay is technically suitable as a human diagnostic. This strengthens the findings of previous studies that used the same targets in a singleplex format [9,16,24], underlining the applicability of these targets for use as human diagnostics. A 100% clinical specificity was obtained when testing nucleic acid extracts from pure cultures from multiple related (non-target) microorganisms that cause fever in Western Europe. However, since other biological agents that could cross-amplify with these TBPs may be present in clinical EDTA samples, it is recommended to repeat the specificity on clinical EDTA blood together with a clinical validation on all TBPs.

A limitation of the developed multiplex qPCRs is that the detection of the included TBPs is based on single targets per TBP, possibly resulting in false negative outcomes due to a (thus far unknown) lack of conservation in the primer and probe annealing sites. To reduce false negative outcomes, the use of multiple targets per TBP could be considered. This strategy would also ameliorate the possibility of contamination from the positive control or PCR amplicons [32]. For this purpose, synthetic positive controls were used that differed genetically from the wildlife sequences. Hence, the amplicon of the positive control could be identified by its size.

Good repeatability was demonstrated in this study for the *A. phagocytophilum* target by exchanging samples with an independent European laboratory. However, due to the scarcity of confirmed patient material, such as blood samples and even DNA extracts of these rare TBD cases, the repeatability of the other targets could not be evaluated [32]. In addition, the occurrence of non-reproducible results [14] and the contamination of positive controls [32] should not be overlooked. Therefore, confirmation of positive test results for these rare diseases by independent (European) laboratories or by sequencing the PCR products is essential in making a reliable diagnosis, at least for the microorganisms that are not yet established as human pathogens or that do not occur regularly (e.g., *Spiroplasma* and *Neoehrlichia*).

Of important notice is that the detection of a microorganism is indicative of infection but not of disease causation and should be complemented by additional diagnostic modalities such as serology and, whenever possible, live culture [14]. An assay that could be of great added value next to the multiplex qPCRs is a multi-pathogen serological diagnostic assay for the combined antigen and antibody detection for multiple TBPs. Although not easily achievable because of the scientific challenge and the time frame required to produce and select all biomaterials needed for the development of such combined assay, it will greatly contribute to the diagnostic field and will be useful for surveillance purposes. Eventually, this combination would allow detecting acute, recent and past infections and thus span the complete diagnostic window for TBP infections. Another important step forward, given the scarcity of the number of available samples of the emerging TBPs in humans, is to set up a ring trial using patient material, as performed in this study for *A. phagocytophilum*. Alternatively, artificial samples (i.e., spiked blood with the DNA target or with DNA from an entire organism) could be used but, however, only when participating laboratories use the same primers and probes. Lastly, validation of the qPCRs for TBP detection in patient materials other than blood is recommended.

## 5. Conclusions

The design of qPCR for the detection of an infectious agent for human diagnosis is usually relatively straightforward. In case of rare or emerging infections such as TBDs, however, the task of designing and evaluating an efficient, sensitive and specific diagnostic modality is a complex task, due to the lack of good human patient material for validation. In this study, three efficient TaqMan-based multiplex qPCR assays were developed for the simultaneous detection of (i) *A. phagocytophilum*, Ca. *N. mikurensis* and *Babesia* spp., (ii) *R. helvetica*, *B. microti* and *S. ixodetis* and (iii) the spotted fever group *Rickettsia* stenos and *Bartonella* spp. The focus of this study was on adding targets for those TBPs that have not yet been established as pathogens or that do not occur regularly, such as Ca. *N. mikurensis*, *R. helvetica* and *S. ixodetis*. The sensitivity and specificity of the three qPCRs was sufficient. However, further clinical validation is still needed, pending the availability of sufficient human samples for testing in different independent laboratories.

## Figures and Tables

**Table 1 microorganisms-10-02222-t001:** Plasmid sequences used for determining the sensitivity of all PCR assays.

Microorganism	MPX I, II or III	Target	Insert Size (bp)	Target Sequences
*A. phagocytophilum*	MPX I	MSP2	85	ATGGAAGGTAGTGTTGGTTATGGTATTATGTTCTGGTGCCAGGGTTGAGCTTGAGATTGGCAGACTACGAGCGCTTCAAGACCAA
*Ca. N. mikurensis*	MPX I	GroEL	105	CCTTGAAAATATAGCAAGATCAGGTAGATGTTCCCTCTACTAATTATTGCTGAAGATGTAGAAGGTGAAGCGCAGACCTTTAGTGCTAAATAAGTTACGTGGTGG
*Babesia* spp. 18S ^1^	MPX I	18S	63	CAGCTTGACGGTAGGGTATTGGCGAGGCAGCAACGGATGTTCTAACGGGGAATTAGGGTTCGA
*R. helvetica*	MPX II	gltA	89	ATGATCCGTTTAGGTTAATAGGCTTCGGTCATGTTCCGATCCACGTGCCGCAGTGCAGACTTGTAAGAGCGGATTGTTTTCTAGCTGTC
*S. ixodetis*	MPX II	rpoB	72	TGTTGGACCAAACGAAGTTGATGTTCGCTAACCGTGCTTTAATGGGATGTTCCCCCAAACACCAATTGTTGG
*B. microti*	MPX II	ITS	88	CTCACACAACGATGAAGGACGCAATGTTCGCAGAATTTAGCAAATCAACAGGATGTTCTCTGAATGTATTGTACACACTGCCTCTGTT
*Bartonella* spp. ^2^	MPX III	ssrA	79	GCTATGGTAATAAATGGACAATGAAATAAATGTTCACCCCGCTTAAACCTGCGACGATGTTCCACCTGGCAACAGAAGC
*Rickettsia* stenos ^3^	MPX III	gltA	84	TCGCAAATGTTCACGGTACTTTATGTTCTGCAATAGCAAGAACCGTAGGCTGGATGGCAGACCACAATGGAAAGAAATGCACGA

The plasmids were determined based on the sequences of the following strains: ^1^ The 18s rRNA gene of *B. divergens*, *B. capreoli*, *B. venatorum*, *B. bigemina*, *B. gibsoni*, *B. canis*. ^2^
*B. alsatica* (IBS 382), *B. bacilliformis* (KC584), *B. birtlesii* (IBS 325), *B. bovis* (91-4), *B. capreoli* (WY-Elk), *B. chomelii* (A828), *B. clarridgeiae* (Houston-2), *B. doshiae* (R18), *B. elizabethae* (F9251), *B. henselae* (Houston-1), *B. grahamii* (V2), *B. japonic* (Fuji 18-1T), *B. koehlerae* (C-29), *B. melophagi* (K-2C), *B. phoceensis* (16120), *B. quintana* (Fuller), *B. rochalimae* (BMGH), *B. schoenbuchensis* (R1), *B. silvatica* (Fuji 23-1T), *B. tamiae* (Th307, Th239, and Th339), *B. taylorii* (M16), *B. tribocorum* (IBS 506), *B. vinsonii* subsp. *arupensis* (OK 94-513), *B. vinsonii* subsp. *vinsonii* (Baker), *B. washoensis* (Sb944nv), and *Bartonella* isolates (Sh6397ga, Sh6396ga, Sh6537ga, Sh8784ga, Sh8200ga, and Sh8776ga). **^3^** Rickettsial members of the spotted fever and typhi group; *R. africae*, *R. aeschlimannii*, *R. heilongjiangensis*, *R. felis*, *R. helvetica*, *R. prowazekii*, *R. typhi*, *R. canadensis*, *R. akari*, *R. australis*, *R. conorii*, *R. honei*, *R. marmionii*, *R. sibirica*, *R. rickettsii*, *R. typhi*, and *R. prowazekii*.

**Table 2 microorganisms-10-02222-t002:** Primer pairs and probes used in this study for the validation of the multiplex qPCR assays. In order to differentiate between the different TBPs within the same multiplex assay, the 5′-end Atto425 probe dye labels were replaced by FAMTM, VICTM and TexRedTM probe dye labels in the multiplex assays.

Microorganism	MPX I, II or III	Target Gene	Primer/Probe	Primer/Probe Sequences (5′-3′)	Ref
*A. phagocytophilum*	MPX I	msp2	ApMSP2F	ATG GAA GGT AGT GTT GGT TAT GGT ATT	[26]
			ApMSP2R	TTG GTC TTG AAG CGC TCG TA	
			ApMSP2P	VIC^TM^-TGG TGC CAG GGT TGA GCT TGA GAT TG-BHQ1	
*Ca. N. mikurensis*	MPX I	groEL	GroEL-F2a	CCT TGA AAA TAT AGC AAG ATC AGG TAG	[5]
			GroEL-R2a	CCA CCA CGT AAC TTA TTT AGC ACT AAA G	
			GroEL-P2a	TexRed^TM^-CCT CTA CTA ATT ATT GCT GAA GAT GTA GAA GGT GAA GC-BHQ2	
*Babesia* spp. (18S) ^1^	MPX I	rRNA	Bab_18SrRNA-F_2016	CAG CTT GAC GGT AGG GTA TTG G	[2]
			Bab_18SrRNA-R_2016	TCG AAC CCT AAT TCC CCG TTA	
			Bab_18SrRNA-P_2016	FAM^TM^-CGA GGC AGC AAC GG-MGB-BHQ2	
*R. helvetica*	MPX II	gltA	Rick_HelvgltA_F2	ATG ATC CGT TTA GGT TAA TAG GCT TCG GTC	[7]
			Rick_HelvgltA_R2	TTG TAA GAG CGG ATT GTT TTC TAG CTG TC	
			Rick_HelvgltA_pr3	FAM^TM^-CGA TCC ACG TGC CGC AGT-BHQ1	
*S. ixodetis*	MPX II	rpoB	Spir_rpoB-F_2016	TGT TGG ACC AAA CGA AGT TG	[27]
			Spir_rpoB-R_2016	CCA ACA ATT GGT GTT TGG GG	
			Spir_rpoB-P_2016	TexRed^TM^-GCT AAC CGT GCT TTA ATG GG-BHQ1	
*Babesia microtii*	MPX II	ITS	Bmicr_ITS_F1_6-2017	CTC ACA CAA CGA TGA AGG ACG CA	[3]
			Bmicr_ITS_R1_6-2017	AAC AGA GGC AGT GTG TAC AAT ACA TTC AGA	
			Bmicr_ITS_Px1_6-2017	VIC^TM^-GCA GAA TTT AGC AAA TCA ACA GG-BHQ1	
*Bartonella* spp. ^2^	MPX III	ssrA	Bart_ssrA-F_2016	GCT ATG GTA ATA AAT GGA CAA TGA AAT AA	[28]
			Bart_ssrA-R_2016	GCT TCT GTT GCC AGG TG	
			Bart_ssrA-P_2016	TexRed^TM^-ACC CCG CTT AAA CCT GCG ACG-BHQ1	
*Rickettsia* stenos ^3^	MPX III	gltA	RickgltA-F-Stenos	TCG CAA ATG TTC ACG GTA CTT T	[8]
			RickgltA-R-Stenos	TCG TGC ATT TCT TTC CAT TGT G	
			Rickglt-probe-stenos	VIC^TM^-TGC AAT AGC AAG AAC CGT AGG CTG GAT G-BHQ1	

The primers and probes were determined based on the sequences of the following strains: ^1^ The 18s rRNA gene of *B. divergens*, *B. capreoli*, *B. venatorum*, *B. bigemina*, *B. gibsoni*, *B. canis*. ^2^
*B. alsatica* (IBS 382), *B. bacilliformis* (KC584), *B. birtlesii* (IBS 325), *B. bovis* (91-4), *B. capreoli* (WY-Elk), *B. chomelii* (A828), *B. clarridgeiae* (Houston-2), *B. doshiae* (R18), *B. elizabethae* (F9251), *B. henselae* (Houston-1), *B. grahamii* (V2), *B. japonic* (Fuji 18-1T), *B. koehlerae* (C-29), *B. melophagi* (K-2C), *B. phoceensis* (16120), *B. quintana* (Fuller), *B. rochalimae* (BMGH), *B. schoenbuchensis* (R1), *B. silvatica* (Fuji 23-1T), *B. tamiae* (Th307, Th239, and Th339), *B. taylorii* (M16), *B. tribocorum* (IBS 506), *B. vinsonii* subsp. *arupensis* (OK 94-513), *B. vinsonii* subsp. *vinsonii* (Baker), *B. washoensis* (Sb944nv), and *Bartonella* isolates (Sh6397ga, Sh6396ga, Sh6537ga, Sh8784ga, Sh8200ga, and Sh8776ga). ^3^ Rickettsial members of the spotted fever and typhi group; *R. africae*, *R. aeschlimannii*, *R. heilongjiangensis*, *R. felis*, *R. helvetica*, *R. prowazekii*, *R. typhi*, *R. canadensis*, *R. akari*, *R. australis*, *R. conorii*, *R. honei*, *R. marmionii*, *R. sibirica*, *R. rickettsii*, *R. typhi*, and *R. prowazekii*.

**Table 3 microorganisms-10-02222-t003:** Evaluation parameters for the three multiplex qPCR assays.

Microorganism	MPX I, MPX II, MPX III	Slope	Efficiency	Dynamic Range C_T_	R^2^	95% LOD (95% CI) ^1^
*A. phagocytophilum*	MPX I	−3.292	101 %	22.79–39.61	0.999	14.67 (11.82–19.76)
*Ca. N. mikurensis*	MPX I	−3.300	101 %	19.65–38.28	0.996	10.79 (9.54–12.69)
*Babesia* spp. 18S	MPX I	−3.359	98 %	19.56–38.65	0.999	33.89 (26.65–47.35)
*R. helvetica*	MPX II	−3.154	108 %	31.66–45.28	0.946	51.41 (41.07–77.13)
*S. ixodetis*	MPX II	−3.022	114 %	17.29–43.82	0.992	41.98 (26.72–106.34)
*B. microti*	MPX II	−3.103	110 %	17.73–41.02	0.992	49.42 (30.13–136.79)
*Bartonella* spp.	MPX III	−3.392	97 %	16.63–40.95	0.994	79.26 (62.56–107.25)
*Rickettsia stenos*	MPX III	−3.447	95 %	16.66–37.80	0.991	11.62 (9.30–16.22)

^1^ [copies/PCR reaction].

**Table 4 microorganisms-10-02222-t004:** Overview after testing the spiked blood to determine the matrix effect per target.

Microorganism	MPX I, MPX II, MPX III	*p*-Values	R^2^
*A. phagocytophilum*	MPX I	0.4190	0.998
*Ca. N. mikurensis*	MPX I	0.1722	0.995
*Babesia* spp. 18S	MPX I	0.1704	0.998
*R. helvetica*	MPX II	0.2838	0.960
*S. ixodetis*	MPX II	0.3312	0.999
*B. microti*	MPX II	0.3641	0.999
*Bartonella* spp.	MPX III	0.2223	0.997
*Rickettsia stenos*	MPX III	0.3321	0.997

**Table 5 microorganisms-10-02222-t005:** Repeatability in two different concentrations of the positive control DNA of the eight qPCRs.

Microorganism	MPX I, MPX II, MPX III	Copy Number/µL	Mean C_T_	SD	%CV
*A. phagocytophilum*	MPX I	10^1^	38.31	0.81	2.11
		10^2^	35.87	0.99	2.77
*Ca. N. mikurensis*	MPX I	10^1^	38.28	0.73	1.90
		10^2^	35.68	0.66	1.84
*Babesia* spp. 18S	MPX I	10^1^	38.65	0.78	2.02
		10^2^	37.59	1.06	2.83
*R. helvetica*	MPX II	10^1^	45.17	2.04	4.51
		10^2^	43.30	1.31	3.03
*S. ixodetis*	MPX II	10^1^	41.00	2.10	5.11
		10^2^	39.36	1.39	3.53
*B. microti*	MPX II	10^1^	41.02	0.61	1.48
		10^2^	39.62	0.67	1.69
*Bartonella* spp.	MPX III	10^1^	40.95	4.72	11.52
		10^2^	37.79	0.80	2.12
*Rickettsia stenos*	MPX III	10^1^	37.80	3.62	9.56
		10^2^	35.78	1.07	2.99

## Data Availability

All the relevant data are within the manuscript.

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
