# Peer review of "Technical Evaluation of qPCR Multiplex Assays for the Detection of Ixodes ricinus-Borne Pathogens"

_microorganisms, 2022, doi:10.3390/microorganisms10112222_

Round 1

Reviewer 1 Report

Tick-borne diseases are on the rise due to a multiplicity of causes, yet suitable diagnostic tests to aid in their diagnosis and prognosis are generally lacking. The authors have taken a step to develop multiplexed qPCR tests to detect and roughly quantify a variety of tick-borne pathogens in diagnostic samples. This study was logically performed, and the manuscript is overall well-written with mostly proper English, the title and abstract are appropriate, and the tables are appropriate and logically laid-out. There remains room for improvement of the manuscript as described below.

1. As the authors themselves point out, these data are all derived from artificial samples and may not behave quite the same with actual clinical samples. They are honest about this in the Discussion. However, it might be appropriate to alter the title to “Preliminary analytical evaluation…” in light of this, as significant work with real samples remains to validate these tests.

2. Sections 2.5.2. and 3.2. Determination of the specificity of the assays on spiked samples is particularly subject to inaccuracy, as other biological agents that may be present (even transiently) in real samples may cross-amplify. The authors should address this possibility in the Discussion rather than giving a false sense of confidence by giving only the statistically-defined specificity, as they have done. Also, is Plasmodium falciparum really a likely problem in most of Europe?

3. line 248. My understanding is the blood was spiked and DNAs extracted, and thus the qPCR was performed on purified DNA. To state that the test was not affected by “..the presence of human blood..” implies there was actual blood contaminating the sample being amplified, which was not the case. Please restate for clarity/ accuracy.

4. The incidence of the TBPs detected by these tests are overall quite low in Europe. It might therefore be useful for the authors to establish the predictive value of positive and negative tests, using a series of spiked samples. I am not asking for this for publication, but even an artificial series would give a better sense of the potential applicability of the tests. If biological pathogen loads are near the LOD and the incidence is less than 1% of the population, the tests will generate many more false negatives than true positives.

5. line 14. The sentence containing “..within the limits of dealing..” is awkward and needs to be reworded. There is no definition for “the limits of dealing”. Instead, “Given the difficulties of dealing with..” might be more appropriate.

6. line 21. Delete “further”, as no clinical validation was performed here.

7. line 29. “hard tick” should be “hard-bodied tick”.

8. line 79. Please clarify how the artificial sequences differ from “wild life” (natural?) sequences.

9. line 213. “..CT-values determined..” should be “..CT-values was determined..”.

10. line 254. Please change “conserveness” to “conservation”.

11. lines 265-269. Using additional criteria besides just a qPCR test (symptoms, history, additional clinical values, etc.) is essential in making a reliable diagnosis, not “highly recommended”.

Author Response

We are pleased to resubmit for publication the revised version of microorganisms-1998739 “Analytical evaluation of qPCR multiplex assays for the detection of Ixodes ricinus borne pathogens.” We sincerely appreciate the constructive criticisms of the reviewer. The corresponding changes and refinements made in the revised paper are summarized and outlined in the attachment. 

Reviewer 2 Report

I read this paper with interest. The following comments may help the authors to improve the manuscript before acceptance.

1.       How is the limit of detection (LOD) calculated? What is the formula for this calculation?

2.       It would be helpful for the readers if the authors present a process scheme from sample preparation to detection.

3.       The limit of detection of the plasmid should be different with the real (clinical) sample. How does the limit of detection of the clinical or real sample calculate? The authors should present a table of comparison.

4.       qPCR technique requires central laboratories. How does the application scale up towards point-of-care? (see, for example, lab-on-a-chip technology: https://pubs.acs.org/doi/abs/10.1021/acs.analchem.9b04863)

5.       The authors should present a table of primers used and where to acquire them.

Author Response

(The authors gave the same response as above.)

Round 2

Reviewer 2 Report

The authors should include the discussion for comment number 2, 3 and 4 in the revised manuscript so that the readers can read them (in case the readers have the same questions as the reviewer.)

Author Response

We are pleased to resubmit for publication the revised version of microorganisms-1998739 “Technical evaluation of qPCR multiplex assays for the detection of Ixodes ricinus borne pathogens.” We sincerely appreciate the constructive criticisms of the reviewer. The corresponding changes and refinements made in the revised paper are summarized and outlined below as agreed with the reviewer and the editor.  
